# RANDOM BUT RIGHT: A GEOMETRIC EXPLANATION FOR EFFICIENT LLM TRAINING

## ABSTRACT

Low-rank gradient optimization for large language models is currently divided into two categories: structured methods that rigorously identify subspaces, and randomized approaches employed primarily for computational efficiency. We question the intuition behind why random projections are effective, tracing this phenomenon to the geometry of the gradient space. Finding that subspace optimization landscape is nearly flat, while a significant portion of gradient information lies outside the core subspace, we introduce GrassWalk and GrassJump, algorithms that navigate the Grassmannian manifold via random walks and jumps. By coupling this randomized exploration with subspace-aware optimizer and recovering the lost gradient signals, we achieve state-of-the-art results. Our findings reframe randomization not merely as a computational shortcut, but as a geometrically principled approach to high-dimensional optimizations.

## 1 INTRODUCTION

Leveraging the insight that gradients evolve within a low-dimensional subspace (Gur-Ari et al., 2018; Schneider et al., 2024), GaLore (Zhao et al., 2024) projects gradients into low-rank manifolds to compress optimizer states. This technique has catalyzed a new family of memory-efficient algorithms (Robert et al., 2025; Rajabi et al., 2025; Chen et al., 2025a; Zhu et al., 2025) that bridge the gap between resource constraints and full-rank training. Low-rank gradient techniques generally fall into two categories based on employing structured subspace adjustments or random projections. Structured methods aim to identify the precise gradient subspace via techniques like singular value decomposition (SVD), as in GaLore (Zhao et al., 2024) and similar works (Chen et al., 2025a), or PowerSGD (Robert et al., 2025) and subspace tracking (Rajabi et al., 2025; Liang et al., 2024). In contrast, other approaches rely on random projections, primarily to avoid the computational cost of structured methods (Zhu et al., 2025; Zmushko et al., 2025; Liu et al., 2025; Chen et al., 2025b; Zhao et al., 2025; Mulrooney et al., 2025; He et al., 2025).

Recently, random projections have emerged as efficient alternatives to SVD (Chen et al., 2025b; Liu et al., 2025; Zmushko et al., 2025; Zhu et al., 2025), effectively replacing expensive operations. To date, randomization has been employed primarily for computational efficiency, yet we lack a fundamental understanding of why it works. Analyzing the energy fraction of low-rank projected gradients and the landscape of subspace optimization, we observe that while a low-rank subspace captures most of the gradient energy, its share declines early during training and is markedly smaller in deeper layers; revealing that the orthogonal energy is non-negligible, and an increasing fraction of learning can occur outside the core subspace. Moreover, we observe that the core gradient subspace evolves in a nearly flat curvature, underscoring the need to account for this structure while adjusting the projection. This perspective for the first time clarifies why certain random-based methods succeed, and also highlights opportunities to design more effective and efficient training algorithms.

Based on these insights and through controlled ablations targeting low-rank gradient challenges, we introduce GrassWalk and GrassJump that achieve state-of-the-art (SOTA) performance using random projections. GrassWalk and GrassJump apply random walks and random jumps on the Grassmannian manifold to update the underlying subspace, while simultaneously adapting the optimizer to subspace changes and restoring information lost in projection, before each weight update.

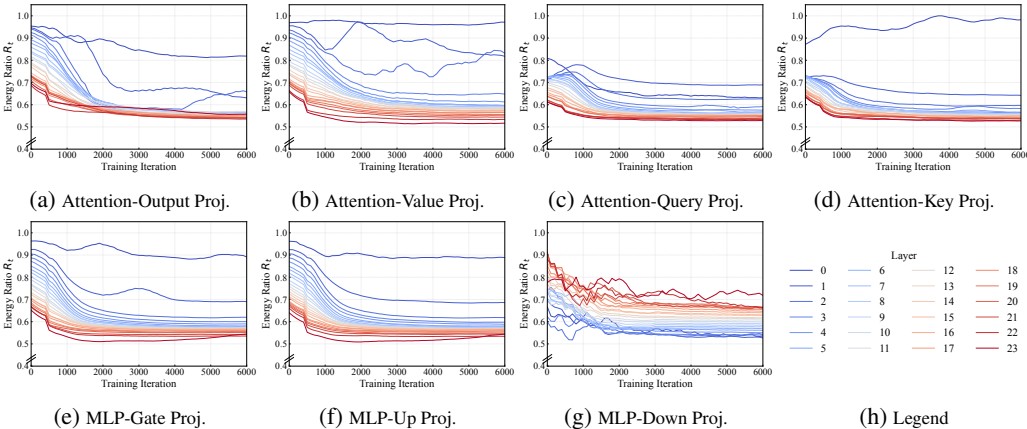

Figure 1: The plots show the fraction of gradient-matrix energy explained by a rank 512 projection across different layer types in a LLaMA-1B architecture. Despite a high lower bound, this fraction declines over training, and deeper layers generally exhibit smaller fractions.

## 2 BEYOND THE CORE SUBSPACE

Low-rank gradient methods reduce memory usage by projecting the gradients $G_t \in \mathbb{R}^{m \times n}$ into a subspace of dimension $r \ll m, n$, as in $\widetilde{G}_t = S_t^\top G_t$, where $S_t \in \mathbb{R}^{m \times r}$ is an orthonormal basis, spanning the subspace ($m \leq n$ without loss of generality). Optimization is performed in this reduced space, and the results are mapped back to the original gradient space for a full-parameter weight update. Also, restricting updates to this core subspace can hinder learning (Song et al., 2025; He et al., 2025), and random projections emerged as an efficient way for updating gradient subspaces (Zhu et al., 2025; Zmushko et al., 2025; Liu et al., 2025; Chen et al., 2025b). However, the intuition behind random projections remains underexplored: what is the role of the core subspace, and if it exists, why can randomized methods get on-par results with structured methods like GaLore (Zhao et al., 2024)? If randomization works, why is there still a performance gap against SOTA baselines?

We pre-trained a LLaMA-1B architecture with 24 decoder layers on the C4 dataset to answer these questions. Also, we adopted the SubTrack++ (Rajabi et al., 2025) setting as a geometrically principled way of adjusting the core subspace, based on the Grassmannian optimization. The Grassmannian $Gr(r, n)$, is the space of all $r$-dimensional subspaces of an $n$-dimensional space. Thus, when we adjust the subspace, we find a point on $Gr(r, n)$ onto which the gradients are projected.

**Low-rank energy fraction.** We quantify the fraction of gradient energy preserved by the low-rank approximation via computing the ratio $R_t = \frac{\|\widetilde{G}_t\|_F}{\|G_t\|_F}$, where the Frobenius norms of the low-rank gradient is divided by the Frobenius norm of the full-rank gradient to calculate this ratio. For a more detailed insight, we clustered results by the type of layers across all 24 decoders. As shown in Figure 1, in all clusters more than 50% of the gradient energy lies in the core subspace; however, the fraction mostly drops in early stages. The decline is most pronounced in deeper layers, where the fraction is consistently lower, suggesting that during pre-training, the subspaces of later layers are less concentrated. This aligns with the common view of LLM pre-training: early layers quickly learn broad, shared features, while continued training shifts capacity toward rarer, more specialized features captured by later layers.

**Subspace optimization landscape.** Prior works have studied the gradient space (Gur-Ari et al., 2018; Zhao et al., 2024; Song et al., 2025), however, up to our knowledge, we are the first to investigate the subspace optimization landscape and its curvature. To this end, at each update step, we compute the derivative of the subspace estimation loss with respect to the current projection matrix. Note that this loss function is not the network training loss, and it minimizes the gradient estimation error caused by the low-rank projection to specify the update direction to adjust the subspace.

$$F(S_t) = \min_A \|S_t A - G_t\|_F^2, \tag{1}$$

For every layer type in the LLaMA-1B architecture, we extract the top 20 singular values of this matrix and report the maximum $i$-th singular value across all 24 layers within each cluster (e.g., the

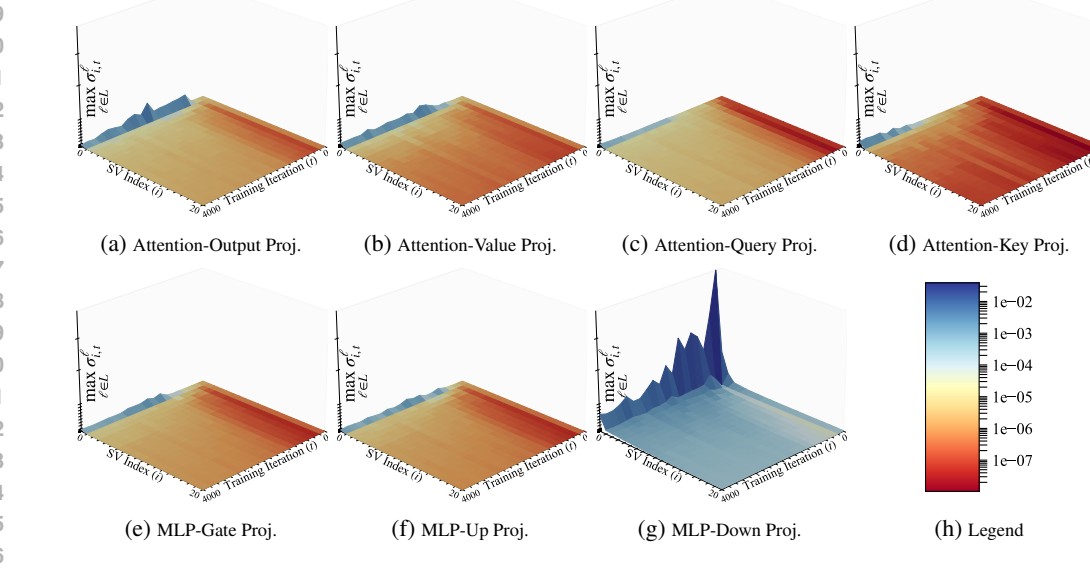

(a) Attention-Output Proj.    (b) Attention-Value Proj.    (c) Attention-Query Proj.    (d) Attention-Key Proj.

(e) MLP-Gate Proj.    (f) MLP-Up Proj.    (g) MLP-Down Proj.    (h) Legend

Figure 2: Top 20 singular values of the subspace estimation error derivative across different layer types. Each plot shows the maximum $i$-th singular value within a given layer type (aggregated across all 24 decoder layers) as training progresses.

largest value among the thirds singular values of query projection layers), providing an upper-bound distribution. Figure 2 shows how the distribution of these singular values evolves during training. Notably, even in the MLP down-projection (Figure 2-g), where the largest singular values appear, the range remains small and decays rapidly. For other layer types, the largest singular values are extremely small, suggesting a flat landscape from the early stages of the training.

These analyses clarify why random projections are effective: The gradient subspace evolve in a flat landscape, a topology where random exploration is essential for efficient optimization, while the considerable portion of energy carried by the orthogonal component of the gradient underscores the necessity to explore outside the core.

**Systematic Ablation.** To compare the effectiveness of subspace update techniques, we consider: **(a) Grassmannian subspace tracking (GST)** from SubTrack++ (Rajabi et al., 2025), tracking the subspace by minimizing a projection error and moving along a Grassmannian geodesic (see their paper for more details). **(b) Random walk**, starting from the low-rank SVD of $G_0$, and then taking random walks on the Grassmannian; **(c) Random projections**, assigning a fresh random projection at each subspace update step; and **(d) SVD-based updates** as in GaLore (Zhao et al., 2024).

For investigating whether the SOTA methods like SubTrack++ (Rajabi et al., 2025) or LDAdam (Robert et al., 2025) are gaining their performance through their subspace adjustment techniques, we are targeting two other challenges of low-rank methods: **(a) Momentum misalignment:** Optimizers like AdamW assume a fixed coordinate system, leading to misalignment when subspaces update. Several works have targeted this issue (Xiao et al., 2025; Zmushko et al., 2025), and we follow the approach used by SubTrack++ (Rajabi et al., 2025) and LDAdam (Robert et al., 2025), called Adaptive Optimizer (AO). This reformulates Adam's states as statistical estimators and rotates them onto the new basis to ensure optimizer remains aligned with the updated subspace. **(b) Information loss:** Low-rank projections discard components orthogonal to the chosen subspace. To recover these signals, we leverage the Recovery Scaling (RS) technique used in FiRA (Chen et al., 2025a) and SubTrack++. RS exploits the consistent scaling factors between dominant and residual subspaces to rescale discarded components. A similar recovery logic is also employed in APOLLO (Zhu et al., 2025). Full technical derivations for the AO and RS components are provided in Appendix A.

To isolate the effect of each component, we subsequently incorporate AO and RS both individually and jointly into each subspace update method, and report the resulting evaluation loss under matched training and evaluation conditions across all configurations. As shown in Table 1, Grassmannian subspace tracking achieves the lowest loss among the update rules when neither AO nor RS is applied, demonstrating the effectiveness of structured manifold-based updates, particularly in

comparison to SVD. This result underscores the limitations of SVD, which is known to be sensitive to noise (He et al., 2025; Rajabi et al., 2025) and to discard prior information (Rajabi et al., 2025).

Adding AO yields larger improvement compared to RS in nearly all settings, with the notable exception of **random projections**. We attribute this to the extent to which each method preserves the core gradient subspace. Grassmannian updates modify the previously learned subspace via controlled rank-1 rotations; whether in subspace tracing or a random walk the divergence remains small. SVD explicitly captures dominant directions; although it is susceptible to noise, the resulting projec-

Table 1: Ablation study on subspace update methods with adaptive optimizer (AO) and recovery scaling (RS). We report evaluation loss (↓) on LLaMA-1B pretraining.

| Update Method | Baseline | +AO | +RS | +AO+RS |
|---|---|---|---|---|
| GST | **5.97** | 4.74 | 5.06 | 3.89 |
| Random Walk | 6.18 | **4.66** | 5.06 | **3.86** |
| Random Projections | 6.16 | 4.94 | **4.89** | 3.87 |
| SVD | 6.17 | 4.79 | 5.30 | 4.00 |

tion still retains most of the informative components. In contrast, random projections select arbitrary subspaces that may discard salient signal. Consequently, RS plays a more critical role in this setting, as the discarded information is more likely to be essential, making its recovery significantly more beneficial. By contrast, the weaker performance of RS without AO is expected. RS relies on the scale factors computed by the Adam optimizer; and if the optimizer is not informed of changes, these scales are corrupted by outdated bases and fail to reflect the intended column-wise rescaling.

## 3 GRASSWALK AND GRASSJUMP

Our analysis motivates using random projections for two key benefits: exploration and generalization. Since significant gradient energy resides in the orthogonal space, random projections force exploration beyond dominant features. Additionally, in a flat curvature, these projections can enhance generalization. Our methods, GrassJump and GrassWalk, are directly inspired by these advantages.

In GrassWalk, we initialize the subspace via the SVD of the first gradient matrix. Then the subspace is updated every $T$ iteration; we then employ the exponential map on the Grassmannian manifold in equation 2 (Bendokat et al., 2024), for a random walk on the manifold. For that, we sample a random matrix $\mathbb{X} \in \mathbb{R}^{m \times r}$ to specify the update direction, and equation 2 requires its SVD to move along the corresponding geodesic. We approximate this decomposition using randomized SVD to reduce computational cost, denoting the result as $\widehat{U}_X \widehat{\Sigma}_X \widehat{V}_X^\top$, with $\tau$ as the update step size.

$$S_{t+1}(\tau) = S_t \widehat{V}_X \cos{(\widehat{\Sigma}_X \tau)} \widehat{V}_X^\top + \widehat{U}_X \sin{(\widehat{\Sigma}_X \tau)} \widehat{V}_X^\top + S_t (I - \widehat{V}_X \widehat{V}_X^\top) \quad (2)$$

In GrassJump, we adopt fully random projection matrices, jumping from one point on the Grassmannian to another every $T$ iterations. At each update, we generate a random Gaussian matrix that can almost preserve orthogonality, especially when the dimensions are large. This approach yields fine-grained random projections, in contrast to block-wise or column-wise subspace selection.

Both of these methods incorporate the other two essential components: Adaptive Optimizer and Recovery Scaling. By leveraging the AO and RS as detailed in Appendix A, they properly adapt optimizer states during subspace shifts and reintroduce the orthogonal signal typically discarded by low-rank projections. The importance of these two components is also validated in Table 1. Also Table 2 compares the performance of our randomized method with other SOTA baselines.

Table 2: Comparison of low-rank methods on pretraining LLaMA-1B model. GrassWalk and GrassJump achieve SOTA results, and the wall-clock time of our randomized methods competes with fastest baselines, such as APOLLO (Zhu et al., 2025), FRUGAL (Zmushko et al., 2025) and SubTrack++ (Rajabi et al., 2025).

| Method | Eval. Loss | Peak Mem. (GB) | Wall Time (m) |
|---|---|---|---|
| AdamW [Full-Rank] | 4.10 | 35.2 | 417.0 |
| GaLore (Zhao et al., 2024) | 6.17 | 31.1 | 522.2 |
| APOLLO (Zhu et al., 2025) | 5.71 | 35.5 | 410.5 |
| LDAdam (Robert et al., 2025) | 4.10 | 34.9 | 532.8 |
| FRUGAL (Zmushko et al., 2025) | 4.22 | 39.3 | 405.1 |
| SubTrack++ (Rajabi et al., 2025) | 3.89 | 32.6 | 429.2 |
| GrassWalk [Ours] | **3.86** | 32.0 | 418.6 |
| GrassJump [Ours] | 3.87 | 32.1 | 415.2 |

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

## A    ADAPTIVE OPTIMIZER AND RECOVERY SCALING

**Adaptive Optimizer.** In Adam, the momentum update rules compute weighted averages of the first- and second-order gradient moments using the parameters $\beta_1$ and $\beta_2$, as shown in the following equations.

$$M_t \leftarrow \beta_1 \cdot M_{t-1} + (1 - \beta_1) \cdot \widetilde{G}_t \tag{3}$$

$$\mathcal{V}_t \leftarrow \beta_2 \cdot \mathcal{V}_{t-1} + (1 - \beta_2) \cdot \widetilde{G}_t^2 \tag{4}$$

When the subspace is updated, we rotate Adam's moments onto the new basis so that the optimizer remains aligned with the updated subspace. Orthogonal projection works well for the first moment but not for the second, since Adam involves nonlinear operations. To handle this, we treat Adam's states as statistical estimates of the first and second moments of each gradient coordinate, and thus using equation 5 and equation 6 for our adaptive optimizer (AO). A similar perspective has also been adopted in prior state-of-the-art methods (Robert et al., 2025; Rajabi et al., 2025).

$$M_t \leftarrow \beta_1(S_t^\top S_{t-1} M_{t-1}) + (1 - \beta_1)\widetilde{G}_t \tag{5}$$

$$V_t \leftarrow \beta_2 \left[ (1 - \beta_2^{t-1}) | (S_t^\top S_{t-1})^2 \cdot (V_{t-1} - M_{t-1}^2) + (S_t^\top S_{t-1} \cdot M_{t-1})^2 | \right] + (1 - \beta_2)\widetilde{G}_t^2. \tag{6}$$

**Recovery Scaling.** Based on the observation that the scale ratio between dominant and bulk subspaces is consistent (Zhu et al., 2025; Chen et al., 2025a), we reintroduce this signal by columnwise rescaling of $\Delta_t$ according to the ratio between the optimizer's output $\widetilde{G}_t^O$ and the raw low-rank gradient $\widetilde{G}_t$, as shown in equation 7. This enables the use of stateful optimizer dynamics without storing the full optimizer states. With a growth-rate limiter $\zeta$, we prevent the scaling from diverging. Specifically, if $\|\Lambda_t\|/\|\Lambda_{t-1}\| > \zeta$, we rescale as per equation 8.

$$\phi_t(G_t)_i = \frac{\|\tilde{G}_{t,:,i}^O\|}{\|\tilde{G}_{t,:,i}\|}, \qquad \Lambda_t = \phi_t(G_t) \, \Delta_t, \tag{7}$$

$$\Lambda_t \leftarrow \Lambda_t \cdot \frac{\zeta \|\Lambda_{t-1}\|}{\|\Lambda_t\|}. \tag{8}$$

Several works (Rajabi et al., 2025; Zmushko et al., 2025; Chen et al., 2025a; Zhu et al., 2025; Robert et al., 2025) have employed various recovery scaling (RS) techniques, and we found this structure to be the most effective complement to our fine-grained random projection matrices.

## B    PRE-TRAINING HYPERPARAMETERS

The hyperparameters of the experiments are reported in Table 3. We evaluated multiple baselines during pre-training LLaMA-1B architecture for 10K steps. All experiments are conducted on a single A6000 GPU, and the final evaluation loss is reported. Method-specific parameters are set based on reported configurations in their original papers, with the subspace updated every 100 steps. Finally, Hyperparameters are reported in Appendix B.

Table 3: Hyperparameters of pre-training Llama-1B architecture.

| Architectural Parameters | Hidden | 2048 |
|---|---|---|
|  | Intermediate | 5461 |
|  | Heads | 24 |
|  | Layers | 32 |
| Shared Parameters | Learning Rate | 1e-4 |
|  | Batch Size | 32 |
|  | Gradient Accumulation | 2 |
|  | Iterations | 10k |
|  | Gradient Clipping | 1.0 |
|  | Warmup Steps | 1000 |
|  | scale | 0.25 |
|  | dtype | bfloat16 |
| Low-Rank Optimizer Methods Parameters | Rank | 512 |
|  | Subspace Update Interval | 100 |
|  | Step-Size | 10000 |

