# OpenReview forum: "Random but Right: A Geometric Explanation for Efficient LLM Training"
_ICLR.cc/2026/Workshop/GRaM — ICLR 2026 Workshop GRaM Poster_

### Official Review · Reviewer_DwrV · 2026-02-19

**Rating:** 4
**Confidence:** 2

**Review:**

Overview:
This paper observes that the eigenvalues of the gradient of the gradient-loss approximation with respect to the projection matrix are small. This helps explain why randomized methods work.
The authors adopt random walks and jumps on the Grassmann manifold to adjust the gradient, and early experiments show promising results.

**Strengths**:
* The identification of a flat loss landscape is a good finding.
* The ablation study is very comprehensive; it divides the subspace update methods into AO and RS.
* The experiments on adapting random walks on the Grassmannian manifold are well conducted.

**Weakness**:
The presentation of the paper is quite poor. It does not flow well and reads more like disconnected pieces of information and findings pasted together.
1. The definitions of core subspace, subspace update, and update within the core subspace are very unclear. These concepts should be presented more clearly.
2. The paper raises the question of the role of the core subspace but does not clearly answer it. Instead, it directly moves to adopting Grassmannian optimization to update the subspace.
3. What is the relationship between the decreasing low-rank energy fraction during training and subspace updates—specifically in relation to your method? From the paper, this only seems to motivate RS, but the connection is not clearly explained.
4. The detailed setup for Figure 2 is missing. Which projection matrix S_t did you compute the gradient with respect to?
5. Is there further motivation on Grasswalk and Grassjump besides the landscape is flat?



**Relevance to topics listed in GRaM call for papers:** Yes

**Originality and novelty:** Yes

**Technical soundness of method:** Yes

**Clarity in writing and organization of the paper** Presentation can be improved

**For the Proceedings track:** N/A

**Double-blind reviewing:** No violations of anonymity were found.

**Use of LLMs:** The text is technical and precise; there are no signs of excessive or improper LLM generation.

**Pmlr Suitability:**

NA

---

### Official Review · Reviewer_i5aJ · 2026-02-20
**Submission 114: New insights to random projections for low-rank learning**

**Rating:** 4
**Confidence:** 4

**Review:**

Overview:
The paper investigates efficient ways to update parameters when low-rank gradients exist (e.g. during LLM finetuning).

PROS:
1. This paper proposes a way to alleviate the SVD computation of gradients (which is done in GaLore) by randomised projection along the Grassmannian manifold.
2. Proposes two methods, GrassWalk and GrassJump.
3. Figure 2 and Table 2 effectively validate the authors' point; the gradient subspaces are generally flat throughout training (fig 2) and the proposed algorithms are faster than GaLore with lower eval loss  (Table 2).

CONS:
1. Title is a bit misleading: as this investigation sheds light on "when random projections are effective" (during low rank and flat curvature gradients), which happen only during fine-tuning or transfer learning, not during pre-training and post-training, writing as "LLM Training" could potentially confuse readers on the pre-training side.
2. To prove Point 1, check the values in Table 1, which suggests that Random Walk (6.18) performs worse than Random Projections (6.16). Although +AO alleviated the problem, +RS exhibits the same problem again, meaning that, however flat the curvature is, it is still important in pre-training. If the paper's claims are correct, then: "Grassmannian walks should outperform arbitrary projections"!
3. I have a question (to reconfirm my point 2): Did you use a shallow 24-layer LLaMA or a pretrained one?

SUGGESTIONS:
1. I think the paper's direction is good, but it is applied to the wrong setup. It would be better to choose a pretrained LLaMA and reproduce Figure 2 and Table 2 for a transfer-learning problem which suits the purpose and will prove the author's point!

Relevance to topics listed in GRaM call for papers: Yes

Originality and novelty: Yes

Technical soundness of method: Yes

Clarity in writing and organization of the paper: Idea is good, but experiments need reevaluation

For the Proceedings track: N/A

Double-blind reviewing: No violations of anonymity were found.

Use of LLMs: The text is technical and precise; there are no signs of excessive or improper LLM generation.

**Pmlr Suitability:**

NA

---

### Meta-Review · Area_Chair_2ksY · 2026-02-23

**Decision:**

Accept

**Metareview:**

The authors identify flat subspaces for gradient methods applied to LLMs fine-tuning, using ideas from the geometry of Grassmannians. The paper is relevant and interesting, but I strongly recommend that the authors take the reviewers' concerns into account for the next version of their paper.

**Relevance To Proceedings:**

Tiny paper — does not apply

**Relevance To Workshop:**

Yes — suitable for GRaM

---

### Decision · Program_Chairs · 2026-03-02

Accept (Poster)